# Measuring Transverse Relaxation with a Single-Beam 894 nm VCSEL for Cs-Xe NMR Gyroscope Miniaturization

**DOI:** 10.3390/s24175692

**Published:** 2024-09-01

**Authors:** Qingyang Zhao, Ruochen Zhang, Hua Liu

**Affiliations:** School of Electronic Information and Electrical Engineering, Shanghai Jiao Tong University, Shanghai 200240, China; zhao_br44@sjtu.edu.cn (Q.Z.); zrc2001@sjtu.edu.cn (R.Z.)

**Keywords:** nuclear magnetic resonance gyroscope, vertical-cavity surface-emitting laser (VCSEL), parametric resonance magnetometer, transverse relaxation

## Abstract

The spin-exchange-pumped nuclear magnetic resonance gyroscope (NMRG) is a pivotal tool in quantum navigation. The transverse relaxation of atoms critically impacts the NMRG’s performance parameters and is essential for judging normal operation. Conventional methods for measuring transverse relaxation typically use dual beams, which involves complex optical path and frequency stabilization systems, thereby complicating miniaturization and integration. This paper proposes a method to construct a ^133^Cs parametric resonance magnetometer using a single-beam vertical-cavity surface-emitting laser (VCSEL) to measure the transverse relaxation of ^129^Xe and ^131^Xe. Based on this method, the volume of the gyroscope probe is significantly reduced to 50 cm^3^. Experimental results demonstrate that the constructed Cs-Xe NMRG can achieve a transverse relaxation time (*T*_2_) of 8.1 s under static conditions. Within the cell temperature range of 70 °C to 110 °C, *T*_2_ decreases with increasing temperature, while the signal amplitude inversely increases. The research lays the foundation for continuous measurement operations of miniaturized NMRGs.

## 1. Introduction

Gyroscopes are essential sensors in inertial navigation systems, used to measure the rotation angle or angular velocity of an object relative to inertial space. With the rapid development of optical measurement technology and quantum sensing technologies, nuclear magnetic resonance gyroscopes (NMRG) have been applied to inertial rotation measurement [1,2,3,4]. This innovative atomic sensor uses alkali metal atoms and noble gas atoms as working substances, detecting the rotation of the carrier system based on changes in the Larmor precession frequency of atomic nuclei in an applied magnetic field [5]. Compared with gyroscopes using other technologies, an NMRG avoids issues of measurement sensitivity to acceleration [6] and has the potential for miniaturization, high precision, and low power consumption [7]. Currently, there are multiple configurations of NMRG [8,9,10,11]; however, they generally utilize alkali metal magnetometers to measure the precession of noble gas nuclei [12,13].

The time required for the transverse macroscopic magnetic moment of noble gas atoms to decay to 1/e (about 37%) of its original value is called the transverse relaxation time (*T*_2_) [14]. Generally, *T*_2_ is positively correlated with the system’s sensitivity and negatively correlated with the linear range [15]. Therefore, measuring the *T*_2_ of noble gas atoms is crucial for improving the performance of an NMRG. Furthermore, successful measurement of the transverse relaxation signal is an important criterion for determining whether the NMRG can operate normally.

The *T*_2_ is typically measured using the free induction decay (FID) method [16]. Pumping the atoms requires a beam of light with a specific wavelength, and the atoms’ precession can be monitored optically [17,18]. Consequently, in traditional FID experiments, two beams of light are often used: one for pumping and one for detection [19]. However, the research of Bell and Bloom [20] has shown that both pumping and detection can be achieved with a single beam of light. We know that using crossed beams or beam splitters increases the complexity of the system. Although a few studies have succeeded in reducing the volume of an NMRG to less than 10 cm^3^ [4,13,21], this achievement is a result of complex processing technology and high manufacturing costs, and it still relies on a dual-beam configuration. However, the advantage of a single-beam configuration is that it can significantly simplify the optical path, making the sensor easier to miniaturize and integrate without increasing the manufacturing process difficulty. Furthermore, it avoids the extra control devices and power consumption introduced by the second beam of light. This method has significant development potential, and both Donley [22] and Haerle [23] have demonstrated its feasibility.

Additionally, traditional NMRG devices typically use external cavity diode lasers (ECDL) or distributed Bragg reflector lasers (DBR) as light sources. The former is relatively large in size, while the latter consumes large amounts of power and has a slow tuning speed. Moreover, both types require bulky frequency stabilization systems. Our goal is to mitigate these drawbacks as much as possible. Compared to the aforementioned lasers, vertical-cavity surface-emitting lasers (VCSEL) have smaller packages and lower power consumption, which can enhance the integration and portability of a compact NMRG [24]. However, their wider linewidth necessitates wavelength locking when used for optical pumping [25]. We adapted the method proposed by Kitching et al. [26] to design a simple and compact VCSEL frequency stabilization system for FID experiments.

In this paper, we propose a miniaturized method for a Cs-Xe NMRG using a single-beam VCSEL to successfully measure the transverse relaxation of Xe. This method significantly reduces the complexity of the relaxation measurement system, decreases the volume of the NMRG probe to 50 cm^3^, and has the potential for further applications in continuous measurements. This method requires the use of a parametric resonant magnetometer and a VCSEL frequency stabilization system, and its basic principle will be described in detail in Section 2. Section 3 covers the structural composition of the NMRG used in the experiment, along with the specific experimental steps. Subsequently, Section 4 presents the experimental results of measuring transverse relaxation. The free induction decay signals of ^129^Xe and ^131^Xe were successfully measured, and the effects of cell temperature and current on the transverse relaxation time and signal amplitude were analyzed. This indicates that the NMRG we constructed has preliminarily achieved static operation, further confirming the effectiveness of the miniaturization method.

## 2. Principle and Methods

### 2.1. Parametric Resonance Magnetometer

When the nuclear magnetic moment of noble gas atoms precesses, it generates a weak alternating magnetic field that is difficult to measure with ordinary magnetometers. The parametric resonance magnetometer (PRM), first proposed by Cohen-Tannoudji et al. [27], is an in situ alkali metal atomic magnetometer. It operates based on the amplification effect of the Fermi contact on the magnetic response, which enhances the sensitivity and accuracy of the measurement, making weak magnetic moment signals easier to detect. This capability makes it suitable for detecting the precession of nuclear magnetic moments in an NMRG. In the NMRG constructed in this paper, a PRM composed of ^133^Cs is utilized to detect the magnetic moment of Xe, as shown in Figure 1. The vapor cell, containing cesium and xenon, is heated to an appropriate temperature. A circularly polarized light beam of a specific wavelength irradiates the vapor cell along the *x*-axis, with the transmitted light received by a photodiode. This beam serves both to pump the alkali metal atoms and to detect the precession of the noble gas atoms.

The Cs ensemble is optically pumped to form a macroscopic magnetic moment ***M***. The magnetic field to be measured, *B*_0_, is oriented along the z-axis, causing ***M*** to undergo Larmor precession around the z-axis. The entire process can be represented by the Bloch equation [28] as
(1)$$\frac{dM}{dt}=\gamma_{\mathrm{Cs}}M\times B+\frac{1}{\tau }({M^{\prime }}_{0}-M)$$
where $\;\gamma_{\mathrm{Cs}}$ is the gyromagnetic ratio of ^133^Cs, about  2π×3498 Hz/μT. τ is the time constant of optical pumping and relaxation, $\;1/\tau =1/T_{p}+1/T$, $\;T_{p}$ is the optical pumping time, and *T* is the thermal relaxation time.${M^{\prime }}_{0}=M_{0}\cdot \tau /T_{p}$, while ***M***_0_ is the saturation magnetic moment in the zero-field static state. ***B*** is the magnetic field vector at the center of the vapor cell. Substituting $B_{x}=0$,$B_{y}=0$, and $B_{z}=B_{0}+B_{1}\cos\omega t$ into Equation (1), results in
(2)$$\left\{\begin{array}{l} \frac{dM_{x}}{dt}=\gamma_{\mathrm{Cs}}(B_{0}+B_{1}\cos\omega t)M_{y}+\frac{1}{\tau }({M^{\prime }}_{0}-M_{x})\\ \frac{dM_{y}}{dt}=-\gamma_{\mathrm{Cs}}(B_{0}+B_{1}\cos\omega t)M_{x}-\frac{1}{\tau }M_{y}\\ \frac{dM_{z}}{dt}=-\frac{1}{\tau }M_{z} \end{array}\right.$$

Expressing the transverse magnetic moment as $M_{\pm }=M_{x}\pm iM_{y}$, we can further obtain
(3)$$\frac{dM_{\pm }}{dt}=\frac{1}{\tau }({M^{\prime }}_{0}-M_{\pm })\mp i\gamma_{\mathrm{Cs}}(B_{0}+B_{1}\cos\omega t)M_{\pm }$$

This means that the applied alternating magnetic field will induce oscillations in the transverse magnetic moment of Cs at each harmonic *pω*. By introducing the Bessel function $J_{n}$ and defining $\omega_{0}=\gamma_{\mathrm{Cs}}B_{0}$, $\omega_{1}=\gamma_{\mathrm{Cs}}B_{1}$ the steady-state solution of Equation (3) is
(4)$$M_{+}(t)=A_{0}{M^{\prime }}_{0}+{M^{\prime }}_{0}\sum_{p=1}^{\infty }\left(A_{p}e^{ip\omega t}+A_{-p}e^{-ip\omega t}\right)$$
where
(5)$$A_{0}=\sum_{n=-\infty }^{\infty }\frac{J_{n}^{2}(\frac{\omega_{1}}{\omega })}{1+i(\omega_{0}+n\omega )\tau }$$
(6)$$A_{\pm p}=\sum_{n=-\infty }^{+\infty }\frac{J_{n}(\frac{\omega_{1}}{\omega })J_{n\mp p}(\frac{\omega_{1}}{\omega })}{1+i(\omega_{0}+n\omega )\tau }$$

Specifically, when the magnetic field *B*_0_ is close to zero, meaning the resonance order n=0, Equations (5) and (6) become
(7)$$A_{0}^{\left(0\right)}=\frac{J_{0}^{2}(\frac{\omega_{1}}{\omega })}{1+i\omega_{0}\tau }$$
(8)$$A_{\pm p}^{\left(0\right)}=\frac{J_{0}(\frac{\omega_{1}}{\omega })J_{\mp p}(\frac{\omega_{1}}{\omega })}{1+i\omega_{0}\tau }$$

Considering that *M*_x_ is the real part of *M*_±_, and combining with Equation (4), the resonance at each harmonic *pω* in *M*_x_(*t*) can be represented in the following form.

When *p* is odd,
(9)$$\delta M_{x}(t)={M^{\prime }}_{0}J_{0}(\frac{\omega_{1}}{\omega })J_{p}(\frac{\omega_{1}}{\omega })\frac{\omega_{0}\tau }{1+{\left(\omega_{0}\tau \right)}^{2}}\sin p\omega t,$$
and when *p* is even,
(10)$$\delta M_{x}(t)={M^{\prime }}_{0}J_{0}(\frac{\omega_{1}}{\omega })J_{p}(\frac{\omega_{1}}{\omega })\frac{1}{1+{\left(\omega_{0}\tau \right)}^{2}}\cos p\omega t$$

It can be seen that Equations (9) and (10) are the dispersion shape and absorption shape. These curves illustrate the response of the parametric resonance magnetometer to the measured magnetic field. Theoretically, the line width is $2/\gamma_{\mathrm{Cs}}\tau$. For the *M*_y_(*t*) component, the shape of the curve is reversed [29].

The laser light intensity *K*_0_ is related to the light pumping rate $1/T_{p}$, which can be expressed as
(11)$$K=\kappa \frac{1}{T_{p}}$$

κ is a constant related to the density of alkali metal atoms. The intensity of the transmitted light of circularly polarized light after passing through the vapor cell can be approximately expressed as
(12)$$K=K_{0}(1+\frac{M_{x}}{M_{0}})$$

If the noble gas atoms perform Larmor precession about the *z*-axis, then the light sensitive to *M_x_* in the *x*-direction will be modulated to the corresponding frequency *ω*, causing the intensity of the transmitted light to vary with this frequency. This transmitted light is received by a photodiode. Thus, the strength of the magnetic field can be determined from changes in the intensity of the transmitted light. Due to atomic transition rules and spin-exchange pumping, the aforementioned measurement method requires a wavelength-stable laser. Therefore, it is necessary to design a laser frequency stabilization scheme for the Cs-Xe NMRG in this paper; otherwise, a stable macroscopic magnetic moment cannot be obtained. The following section will introduce the principle of VCSEL frequency stabilization applied in this NMRG.

### 2.2. VCSEL Frequency Stabilization

The transition energy levels of the alkali metal atom ^133^Cs exhibit hyperfine structure [30]. To polarize ^133^Cs, the pump laser wavelength must be tuned to the vicinity of a specific transition line. ^133^Cs has two transition lines (the D1 line and the D2 line), each including multiple hyperfine transition lines. Both lines can achieve optical pumping polarization. We chose the D1 absorption line of $6^{2}S_{1/2}\to 6^{2}P_{1/2}$ (approximately 894.6 ± 0.03 nm), as illustrated in Figure 2.

The emission wavelength λ of a VCSEL is influenced by both the injection current and the laser temperature, which can be expressed as
(13)$$\lambda =\frac{2nL}{q}$$
where *n* is the refractive index of the resonant cavity, *L* is the length of the resonant cavity, and *q* is the order of the longitudinal mode. Both *n* and *q* vary with the injection current and the temperature of the laser. Generally, the larger the current and the higher the temperature, the longer the emission wavelength. To address the issue of wavelength drift during long-term operation, it is essential to implement closed-loop control of the wavelength. An effective method is to maintain the temperature of the VCSEL at a fixed point while modulating the injection current of the laser to lock the VCSEL’s wavelength at the optimal absorption point of ^133^Cs. As shown in Figure 3, we control the temperature and injection current of the VCSEL separately in this NMRG and use the information carried in the transmitted light to achieve wavelength locking.

The temperature control method is different from that used in the vapor cell’s oven. It uses a negative temperature coefficient (NTC) thermistor and a thermoelectric cooler (TEC) integrated into the VCSEL’s package to achieve it. The injection current is generated by summing the outputs of a precision DC current source and an AC current source. We use a low-frequency signal (kHz) to modulate the injection current and lock the VCSEL frequency to the absorption peak by adjusting the offset of the phase-locked result. This method follows the design by Kitching et al. [26]. The implementation principle of this frequency stabilization method will be introduced below.

When a weak AC component is added to the VCSEL injection current *I*(*t*), *I*(*t*) can be represented as
(14)$$I(t)=I_{0}+I_{c}\sin\omega_{c}t$$
where *I*_0_ is the DC component, *I*_c_ is the AC modulation amplitude, and *ω*_c_ is the modulation frequency. The light transmitted through the cell is transformed into a voltage signal *U*(*t*) by the photodetection module. Because the amplitude of the modulation signal is very small, *U*(*t*) and *I*(*t*) can be considered to have a linear relationship. Considering the vapor cell’s absorption of light and the modulation of light intensity by the alternating component, *U*(*t*) near *I*_0_ can be expressed as
(15)$$U(t)=U_{0}+k_{c}U_{c}\sin(\omega_{c}t+\varphi )$$

*U*_0_ and *U*_c_ are the voltage values corresponding to the intensity of the light for the direct and alternating components of the laser drive current, respectively. *k*_c_ is the coefficient for the change in light absorption rate of the cell caused by current modulation, and *φ* is the total phase shift of the photonic signal transformation. Equation (15) indicates that the alternating modulation contained at the light emission end will manifest at the receiving end at the same frequency *ω*_c_. To extract the alternating signal from *U*(*t*), specifically the second term, the demodulation is performed using the reference signal $U_{s}\sin\omega_{c}t$, resulting in
(16)$$y(t)=\frac{1}{2}k_{c}U_{c}U_{s}\cos\varphi +\frac{1}{2}k_{c}U_{c}U_{s}\cos(2\omega_{c}t+\varphi )$$

*U*_s_ is the amplitude of the reference signal. After low-pass filtering, *y*(*t*) results in the demodulated signal y:(17)$$y=\frac{1}{2}k_{c}U_{c}U_{s}\cos\varphi$$

Due to the selective characteristics of the vapor cell for specific wavelengths of light, the light absorption effect is most pronounced at the four transition points on the D1 line of ^133^Cs: $F_{g}=3\to F_{e}=4$, $F_{g}=3\to F_{e}=3$, $F_{g}=4\to F_{e}=4$, and $F_{g}=4\to F_{e}=3$. These correspond to the four minima on the absorption curve. At these points, the rate of change of the light absorption rate is zero, meaning $k_{c}=0$, which corresponds to the four zero-crossing points on the demodulation curve where the values transition from negative to positive. Thus, it is possible to stabilize the emission wavelength of the VCSEL by implementing closed-loop control of the injection current, locking the phase-locked output at a specific transition point. For detailed methods, please refer to Section 3.1.2.

## 3. Experiment

### 3.1. Composition of Experimental System

#### 3.1.1. Cs-Xe NMRG

Figure 4 illustrates a schematic diagram of the overall system. The core of the apparatus is a cubic sample vapor cell measuring 4 × 4 × 4 mm^3^, made of high-temperature-resistant glass. The vapor cell is filled with ample ^133^Cs, 5 torr ^129^Xe, 5 torr ^131^Xe, 20 torr N_2_, and 200 torr He as the buffer gas.

The internal structure of the oven consists of multiple layers, as shown in Figure 5b, with the outermost layer made from high-temperature-resistant resin to minimize heat loss from the cell. Internal gaps are filled with materials of low thermal conductivity to further enhance insulation. The heating film is a dual-layer flexible printed circuit (FPC) board, shown in Figure 5c, designed to avoid stray magnetic fields generated by heating and achieved by etching reverse-parallel circuits on the upper and lower copper foils. The FPC supplies high-frequency alternating current to heat the vapor cell, maintaining a temperature of 90 ± 0.05 °C. The oven is centrally located within a five-layer nickel-iron alloy magnetic shielding device, with the external magnetic field applied through a three-dimensional coil. A VCSEL is tuned to the D1 absorption line of ^133^Cs and is converted into parallel left-handed circularly polarized light after passing through a collimator and polarizer. The total volume of the sensor probe is approximately 50 cm^3^.

#### 3.1.2. VCSEL

A beam of light generated by a VCSEL (vertical-cavity surface-emitting laser) is collimated and polarized to illuminate the vapor cell, and it is received by a photodetector, which serves both to pump the alkali atoms and to detect nuclear precession. We use a single-transverse-mode 894 nm infrared VCSEL, with single-linear-polarized emission designed for pulsed and modulated applications, featuring an output power of 360 μW (approximately 20% power loss by the time it reaches the photodetector).

The VCSEL is housed in a TO-46 package. Incorporated within the VCSEL are a thermistor and a thermoelectric cooler (TEC), and a retaining ring is installed externally to adapt to the collimator, as shown in Figure 6a. First, the operating temperature of the VCSEL is stabilized at an appropriate set point using a TEC driver controller and implementing PID closed-loop control, achieving a temperature stability range of ±0.002 °C. A high-precision current source chip is utilized to modulate the injection current, with an alternating signal generated by a direct digital synthesis (DDS) module at approximately 1.2 kHz. The driving and control functions of the VCSEL are implemented on an integrated circuit board with a size of 10 × 6 cm^2^, as depicted in Figure 6b.

The photodetector receives the transmitted light, converting it into an electrical signal that is filtered before being input into the phase-locked amplifier, enabling the extraction of the demodulated signal. We perform scanning over a specified current range and utilize a 24-bit ADC to collect data, resulting in two curves: the light-absorption-mode curve and the demodulation result curve, as shown in Figure 7. The demodulation result is a curve that exhibits a dispersion mode. The four minima of the absorption curve correspond to the four hyperfine level transitions of the ^133^Cs D1 line and to the four zero-crossing points *I*_1_ to *I*_4_ on the demodulation result curve. By performing linear fitting, we approximate that the current coefficient of the VCSEL is 0.6 nm/mA. In the actual experiment, we selected point *I*_4_ ($F_{g}=4\to F_{e}=3$) as the current control point. When the system was powered on for the first time, we performed a rough scan of the current within a certain range to identify our control point. Then, based on the phase-locked output result, the injection current was controlled using digital PID closed-loop control, allowing the emission wavelength of the VCSEL to be locked at the absorption peak.

In addition, we tested this method over a period of time, and Figure 8 shows the results of the closed-loop control. It can be observed that the fluctuations in the output value of the phase-locking module did not exceed ±5 mV. Referring to the demodulation result curve in Figure 7, it is noted that near the *I*_4_ point, the fluctuation of the injected current did not exceed ±4 nA. This means that the estimate of the wavelength fluctuation within 180 s is ±3 × 10^−6^ nm. However, this result is merely an experimental estimate of the wavelength fluctuation range and cannot be regarded as a definitive indicator of wavelength locking; we present it here solely for preliminary discussion. In fact, based on the transverse relaxation signals we successfully measured in Section 4, we believe that this VCSEL frequency stabilization method can meet the current system’s requirements.

### 3.2. Experimental Setup

We utilize a field-switch technique for the FID experiment, which is divided into two phases: optical pumping and optical detection. This experimental method is similar to techniques used by Donley [22] and Haerle [23]. The experimental settings described below are all based on the coordinate system provided in Figure 4.

In the first phase, which is the optical pumping phase, we need to prepare polarized Xe atomic clusters. In our configuration, the propagation direction of the pump beam is along the *x*-axis, and a magnetic field *B_p_* is applied along the *x*-axis during the pumping phase. Consequently, the magnetization of ^133^Cs increases along the *x*-direction. Using the methods and circuits introduced in Section 3.1.2, the VCSEL light source is tuned to the fourth absorption peak of the ^133^Cs D1 line. The spin polarization of Cs is then transferred to the Xe nuclear spins through spin-exchange collisions between Cs and Xe atoms, thereby forming a macroscopic magnetic moment of Xe in the *x*-direction. At the start of the optical detection phase, the field-switching device is used to turn off *B_p_* while activating *B_dc_* and $B_{1}\cos\omega t$, completing the magnetic field switch. Specifically,$B_{1}\cos\omega t$ in the *y*-direction has a frequency of 10 kHz and an amplitude of ~1100 nT; *B_dc_* in the *z*-direction is ~260 nT, and *B_p_* in the *x*-direction is ~1 µT. After the signal completes the photoelectric conversion, it undergoes phase-locked amplification (the reference frequency is the same as the frequency *ω* of the alternating magnetic field) and filtering. Subsequently, the output signal from the lock-in amplifier contains the FID signal, which is collected by the ADC and transmitted to the computer via a serial port, with a collection time set to approximately 30 s. Ultimately, the data are processed and visualized using the computer.

## 4. Experimental Results

Figure 9a shows the time-domain signal of the free induction decay of Xe detected by the ^133^Cs parametric resonance magnetometer. This signal reflects the magnetic field generated by the precession of the Xe atomic nuclei, which is superimposed on the ^133^Cs resonance signal and extracted by the lock-in amplifier, expressed in volts. Figure 9b shows the Fourier transform of the data after the magnetic field switching from Figure 9a, describing the frequency components present in the FID signal of Xe. It can be observed that there is a resonance peak for ^129^Xe at $\omega_{L1}/2\pi =3.08\;\mathrm{Hz}$, which corresponds well with the external static magnetic field we applied, $B_{dc}=260\;\mathrm{nT}$. Furthermore, a weaker resonance line for ^131^Xe is found in the low-frequency range, with $\omega_{L2}/2\pi =0.91\;\mathrm{Hz}$.

The temperature of the cell is a crucial factor influencing the spin-exchange collisions between alkali metal atoms and noble gas atoms, which indirectly determine the polarization rate of the Xe atoms, thereby affecting the relaxation of the Xe atoms. Experimental observations indicate that the transverse relaxation time and signal strength vary at different cell temperatures. Therefore, it is necessary to measure the transverse relaxation of Xe at various cell temperatures, as this will assist us in determining the optimal operating temperature for the NMRG to some extent. Based on the experimental setup outlined in Section 3.2, we measured FID signals across a temperature range of 70 °C to 110 °C while keeping other experimental conditions constant by adjusting the cell temperature. Figure 10 presents the measurement results at temperatures of 80 °C, 90 °C, and 100 °C.

Figure 11 presents the measurement results under different cell temperature conditions, comparing both the transverse relaxation time *T*_2_ and the normalized intensity of the measurement signals. It is evident that in the Cs-Xe NMRG we constructed, *T*_2_ shows a downward trend with an increase in the cell temperature, while the FID signal amplitude displays an opposite trend. At relatively low temperatures, the probability of binary collisions between Cs and Xe decreases [32], and the probability of ternary collisions between Cs, N_2_, and Xe also decreases. Consequently, the relaxation signal is weak, leading to a lower signal-to-noise ratio, despite achieving a longer transverse relaxation time. As the cell temperature increases, the spin-exchange collision effect strengthens, resulting in a higher polarization rate of Xe atoms. In summary, the optimal cell temperature for the NMRG developed in this study is between 85 °C and 95 °C. However, utilizing different compositions and ratios in the vapor cell may yield varying results, which will be explored further in subsequent research. Finally, we preliminarily explored the relationship between *T*_2_ and changes in the injection current. We conducted FID experiments at points *I*_1_, *I*_2_, *I*_3_, and *I*_4_ (the maximum interval is no more than 0.075 mA), obtaining a series of curves as shown in Figure 12. Theoretically, the optical power at different current control points varies, which may lead to changes in the pump rate, thereby affecting the transverse relaxation time of the atoms. However, the experimental results indicate that the measured *T*_2_ exhibits only a small change. We believe this is due to the minimal difference in current values, which leads to negligible changes in optical power. Given that our main experimental objective has been largely achieved and considering the scope of this study, we will not discuss this further.

## 5. Conclusions

Transverse relaxation is a critical criterion for assessing whether an NMRG can operate normally and serves as a prerequisite for continuous operation. Experimental results indicate that we can reliably and straightforwardly measure the transverse relaxation of ^129^Xe and ^131^Xe, enabling discontinuous measurements under static conditions. This also confirms the validity of the gyroscope constructed in this study. Furthermore, this paper conducts a preliminary investigation into the influence of temperature and current on the relaxation effect. By comparing experiments and considering both the signal-to-noise ratio and transverse relaxation time, the optimal operating temperature range for the gyroscope is determined to be 85 °C to 95 °C. We hope that our work will contribute to the miniaturization and continuous measurement operation of this type of NMRG, and further research will be conducted in the future

## Figures and Tables

**Figure 1 sensors-24-05692-f001:**
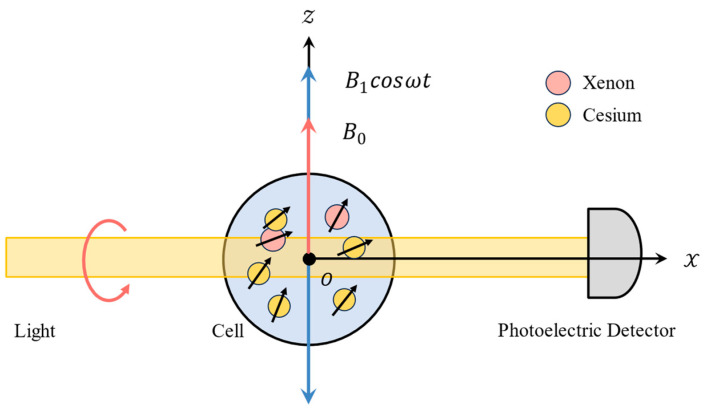
A simplified schematic of ^133^Cs PRM.

**Figure 2 sensors-24-05692-f002:**
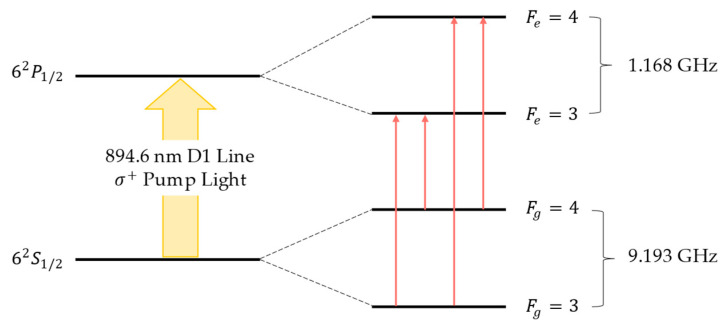
The D1 absorption line of ^133^Cs.

**Figure 3 sensors-24-05692-f003:**
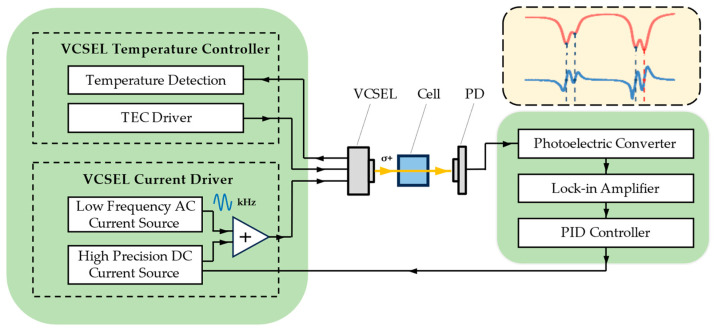
Schematic diagram of the system for achieving wavelength locking of VCSEL in Cs-Xe NMRG. (Parts of the optical path structure and mechanical structure are omitted in the figure.).

**Figure 4 sensors-24-05692-f004:**
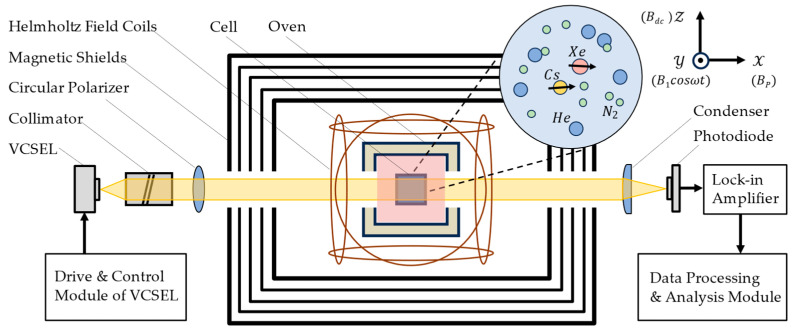
Schematic diagram of the overall experimental system.

**Figure 5 sensors-24-05692-f005:**
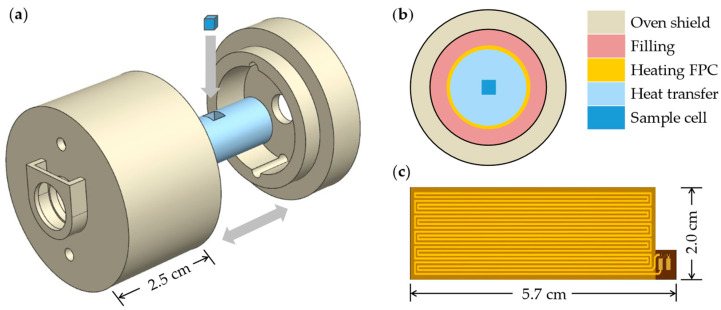
The structure of the probe part of the NMRG. (**a**) Assembly diagram. (**b**) The composition of the layers inside the oven. (**c**) Dual-layer flexible printed circuit heating film.

**Figure 6 sensors-24-05692-f006:**
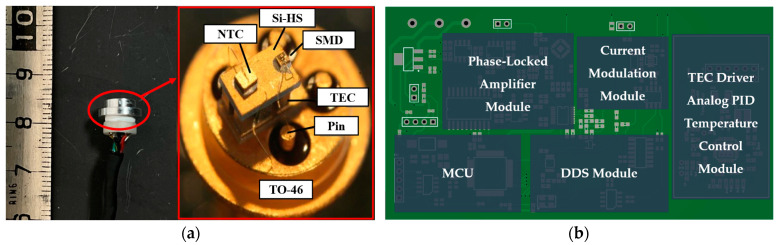
VCSEL and its peripheral circuits. (**a**) The TO-46 package of the VCSEL, with its internal structure (cited from [31]) highlighted in the red box. NTC: negative temperature coefficient thermistor; Si-HS: silicon-based heat sink; SMD: surface mount devices; TEC: thermoelectric cooler. (**b**) The driving and frequency locking circuit board for the VCSEL, size 10 × 6 cm^2^.

**Figure 7 sensors-24-05692-f007:**
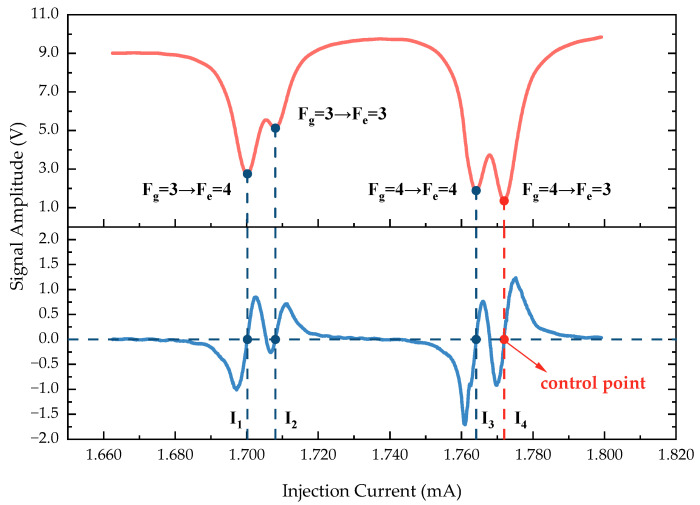
Current scanning curve. (**Top**): absorption curve; (**Bottom**): demodulation result curve.

**Figure 8 sensors-24-05692-f008:**
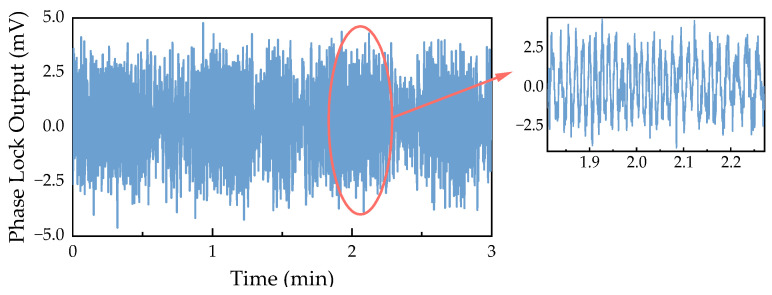
Test results of closed-loop control of VCSEL wavelength within 180 s.

**Figure 9 sensors-24-05692-f009:**
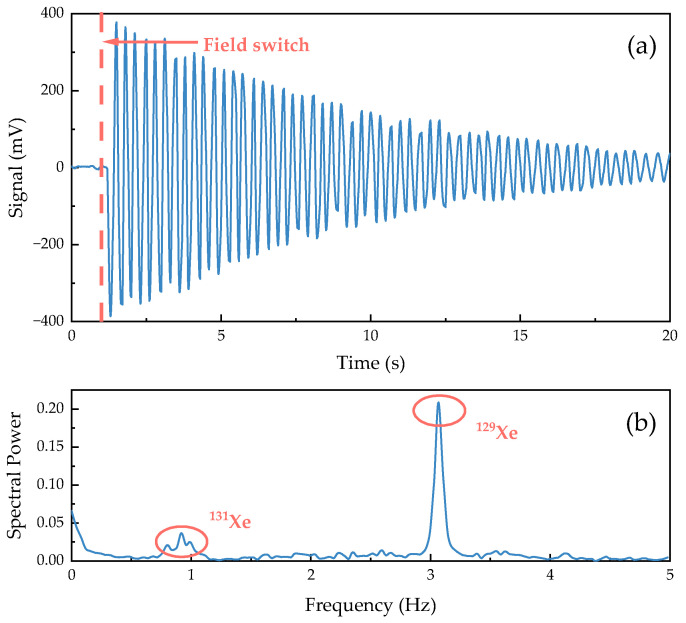
Results of the FID experiment. (**a**) Time domain signal. (**b**) The FFT transform of the time domain signal.

**Figure 10 sensors-24-05692-f010:**
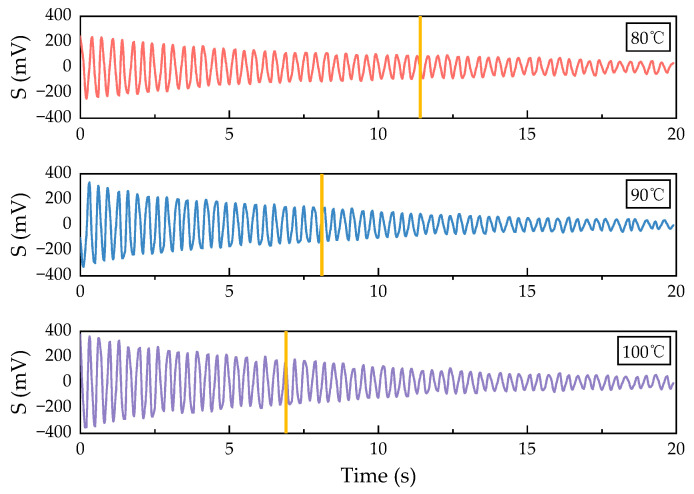
Comparison of FID signals at different cell temperatures.

**Figure 11 sensors-24-05692-f011:**
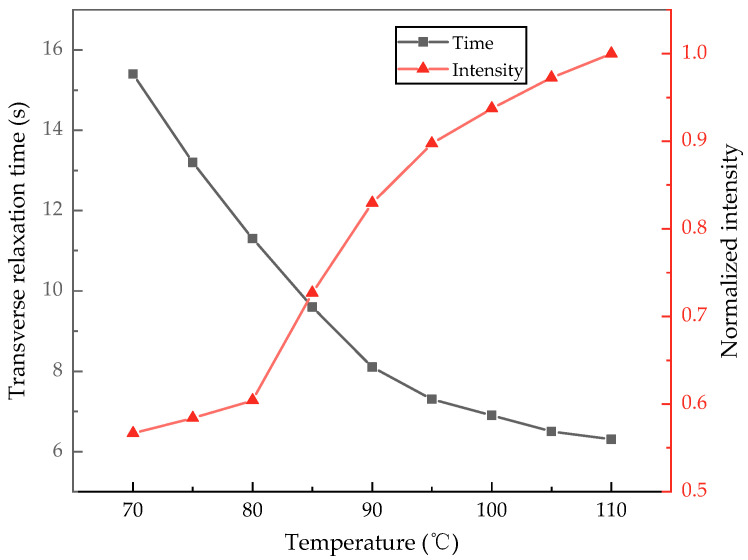
The relationship between *T*_2_ and signal amplitude and cell temperature changes.

**Figure 12 sensors-24-05692-f012:**
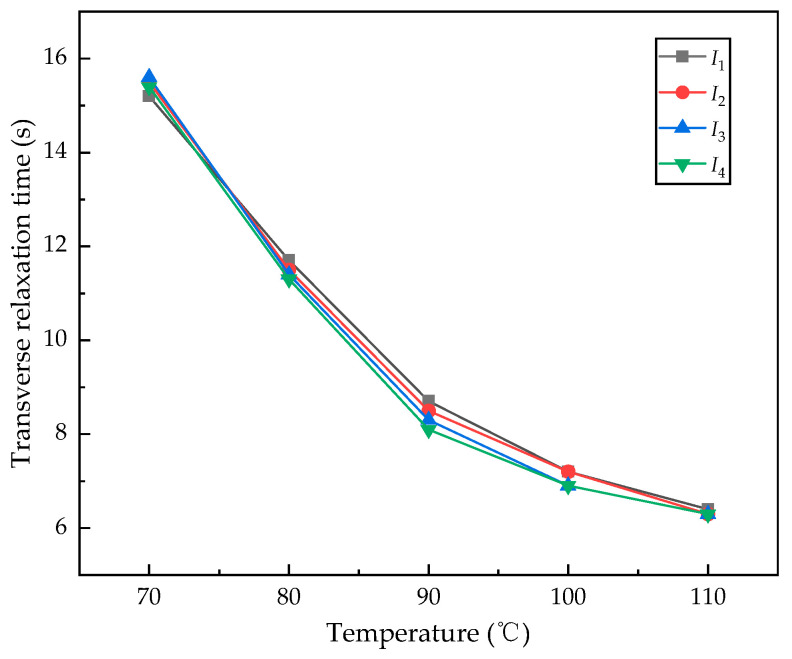
The relationship between *T*_2_ and different injection currents (*I*_1_~*I*_4_).

## Data Availability

Not applicable.

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
