# Peer review of "Measuring Transverse Relaxation with a Single-Beam 894 nm VCSEL for Cs-Xe NMR Gyroscope Miniaturization"

_sensors, 2024, doi:10.3390/s24175692_

Round 1

Reviewer 1 Report

Comments and Suggestions for Authors

Overall contents are well structured and well explained. Some minor errors such as space are needed to be improved. 

Comments on the Quality of English Language

overall contents are well structured and well explained. Some minor errors such as space are needed to be improved. 

Reviewer 2 Report

Comments and Suggestions for Authors

See comments attached.

Comments on the Quality of English Language

The quality of English in this manuscript is fine, only some minor grammatical corrections. 

Reviewer 3 Report

Comments and Suggestions for Authors

This paper presents a novel method utilizing a single-beam VCSEL to construct a 133Cs parametric resonance magnetometer for measuring the transverse relaxation of 129Xe and 131Xe, significantly reducing the gyroscope probe volume to 50 cm3. Furthermore, a laser frequency stabilization method based on injection current modulation is introduced, achieving wavelength fluctuations of less than 3×10-6 nm. These works are of great significance for rapidly measuring atomic transverse relaxation time and improving wavelength stability, therefore, I believe this is a good manuscript. However, there are still some shortcomings, as follows:

1、 Suggest adding other methods to the manuscript to test the transverse relaxation time, or theoretically calculate the transverse time, and compare it with the measurement results in the manuscript to demonstrate the accuracy of the measurement results proposed in this paper.

2、 The transverse relaxation time at different temperatures was tested using a single beam, but it seems that the impact of different light absorption at different temperatures on the measurement results has not been analyzed.

In summary, it is recommended to make major revisions to the article, which would make it a good piece of work.

Reviewer 4 Report

Comments and Suggestions for Authors

The article under consideration is devoted to some aspects of the development of a nuclear gyroscope and the application of its circuit to measure the transverse relaxation time of a nuclear paramagnet (xenon). The experimental part of the work is carried out at a sufficiently high level, and the results are presented clearly.

However, I cannot agree with the main statements made by the authors.

==========

Firstly (and most importantly), the authors write:

L16: “Furthermore, a laser frequency stabilization method based on 16 injection current modulation is introduced, achieving wavelength fluctuations of less than 3×10-6 17 nm”.

At the same time, it follows from the text of the article that the “new method” consists of modulating the laser current and demodulating the radiation that has passed through the cell.

L198: “Thus, it 198 is possible to stabilize the emission wavelength of the VCSEL by implementing closed- loop control of the injection current, locking the phase-locked output at a specific transition point”.

This method has been the main method of frequency stabilization for any diode lasers, including VCSELs, for decades. Below are several (randomly selected from many) publications that use this method:

1.       Gruet, Florian, et al. "A miniature frequency-stabilized VCSEL system emitting at 795 nm based on LTCC modules." Optics and Lasers in Engineering 51.8 (2013): 1023-1027.

2.       Hall, John L., Matthew S. Taubman, and Jun Ye. "Laser stabilization." OSA Handbook v14 (1999).

3.       Serkland, Darwin K., et al. "VCSELs for atomic clocks." Vertical-Cavity Surface-Emitting Lasers X. Vol. 6132. SPIE, 2006.

4.       Al-Samaneh, Ahmed. "VCSELs for Atomic clock demonstrators." Annual report (2013).

As far as I can judge, the entire theoretical part of the work does not contain anything new or original. Section 2.1 contains a description of the Parametric Resonance Magnetometer, first given in the original articles of 1946 (F. Bloch) and 1973 (R. Slocum). Section 2.2 contains a list of well-known principles of diode laser stabilization.

Furthermore, the method of xenon polarization used by the authors remains a complete mystery to me. The discussion is complicated by the fact that in Fig. 1 and Fig. 3 the authors use different coordinate systems. Moreover, in L278 they introduce a new value Bp, without explaining it or showing it in the figure:

L278: “Simultaneously, is applied in the z-direction, leading to spin polarization of Xe, thereby forming a macroscopic magnetic moment (which, in our case, aligns along the z-direction)”.

Thus, this field is used to polarize xenon along the beam. Question: how is macroscopic polarization of xenon achieved if the energy splitting of Xe levels in the field Bp=1 uT is many orders of magnitude smaller than the Boltzmann energy kT?

If spin exchange effects with optically oriented Cs are used in this process, this requires an explanation.

It is also necessary to note the fundamental inaccuracy:

L85: “When the nuclear magnetic moment of noble gas atoms precesses, it generates a weak 85 alternating magnetic field that is difficult to measure with ordinary magnetometers […] an alkali metal atomic magnetometer capable of measuring near zero-field conditions with very high sensitivity and performing in-situ measurements in the cell. This makes it suitable for detecting the precession of nuclear magnetic moments.”

The main thing that makes an alkali metal atomic co-magnetometer suitable for detecting the precession of nuclear magnetic moments is the Fermi contact amplification (appox. 800 times) of the effective magnetic fields created by Xe nuclea.

=============

There are also a few less significant points to note:

1.

L40: “to decay to 37% of its original value”

What is this value - 37%? Do the authors mean 1/e?

2.

L42: Moreover, T2 also determines the angular random walk of the gyroscope, meaning the system's stability is also influenced by T2.

The angular random walk determines the sensitivity of the gyroscope, while its stability is primarily determined by the drifts of its parameters.

3.

L140: “If a weak magnetic field is undergoing Larmor precession”

The field cannot undergo Larmor precession.

4.

L266: “As a result, the fluctuations in the laser wavelength are calculated to 266 be less than…”

The authors do not investigate the potential long-term stability of the method and do not provide any estimates.

5.

L267: “compared to the linewidth of the entire D1 absorption line, which is nm. 2.8 10-2 nm”

If we convert this figure from nanometers to gigahertz, we get approximately 12 GHz. Therefore, the authors compare the accuracy of their stabilization with the full width of the composite D1 line, whereas the comparison should be made with the width of a separate transition (< 1 GHz).

6.

L316: “signal intensity”

The term "signal amplitude" would be more appropriate.

=======

To sum up the above, I cannot recommend the article for publication. I believe that the authors should completely rework it, excluding insufficiently substantiated statements (primarily this concerns the statement about the novelty of the laser stabilization method), and focusing on what has actually been done. At the same time, any statements about the achieved stability values, etc. should be accompanied by assessments of not only short-term noise, but also long-term drifts.

Reviewer 5 Report

Comments and Suggestions for Authors

This is a well-constructed paper that very much fits the remit of the journal. The area covered is very topical looking at the factors needed to optimize the operation of a spin-exchange pumped nuclear magnetic resonance gyroscope (NMRG). This technology is of interest for quantum-based gyroscopes for navigation devices. This NMRG approach could offer more continuous operation. The paper is developed logically with the background theory/principles, the overall experimental set up along with the details of vertical-cavity surface-emitting laser (VCSEL) and then the experimental outputs which are the transverse relaxation and intensity of the xenon signal. Their strong variation with temperature means the temperature has to be well controlled, as well as there being an optimum temperature for operation. It is very interesting to see how the method of pumping the caesium signal and then using the modulation caused by the xenon magnetization to determine its transverse relaxation time. The data looks to have been carefully collected and analysed.

The writing is clear, with very few typographical errors (see some very minor points below). The reference list is well judged in that there are helpful background references and those that relate to the latest results.

Points for Consideration

1.  Specialists will understand all of the acronyms, but to make the paper more immediately accessible to a wider audience they should all be spelt out the first time they are used, e.g. VCSEL, EOM, AOM, etc.

2.  There are some experimentally derived values such as the transverse relaxation time. There should probably be some errors estimated for these values.

Minor typographical corrections

3.  p4, line 129, there needs to be a space -are dispersion

4.  p7, line 241, needs to be a space after the .

Round 2

Reviewer 3 Report

Comments and Suggestions for Authors

1、The author's response to my review comments stating that it is impossible to measure the true value of transverse relaxation time is clearly incorrect. When the gas temperature, pump optical power density, gas composition, etc. are determined, the transverse relaxation time is also determined accordingly.

But the author has provided explanations for all the review comments and supplemented the experiments, so I suggest accepting this article.

Reviewer 4 Report

Comments and Suggestions for Authors

I am grateful to the authors for their detailed responses to all my comments. All the answers, except the first one, satisfied me completely and did not raise any further questions. Unfortunately, the authors' response to Comment #1 could not make me change my mind. The authors write:

“First, references 2 to 4 all focus on research related to achieving VCSEL frequency stabilization in CPT atomic clocks. Due to the stringent interaction conditions required for CPT and the complex frequency counting principles in atomic clocks, the modulation signal for the injected current must be a precise high-frequency (GHz) signal. For example, Reference 4 states, "For atomic clock operation, the bias current of the VCSEL is modulated by a harmonic signal with a frequency of 4.596 GHz, equal to half the Cs hyperfine ground splitting frequency (Figure 1).” This approach presents significant design challenges and can be quite costly.

However, we designed a VCSEL frequency stabilization scheme specifically for the Cs-Xe NMRG structure. Our primary aim is to facilitate the pumping of Cs atoms, and this method does not rely on the CPT effect. As a result, the injected current modulation signal can utilize a low-frequency (kHz) AC signal, which can be easily generated using standard low-cost DDS modules. This approach not only simplifies the implementation but also mitigates the PCB design challenges associated with ultra-high frequency signals”.

The above text shows that the authors' ideas about the principles of operation of quantum sensors are erroneous. Thus, in CPT atomic clocks, modulation at a frequency of 4.596 GHz is used not to stabilize the laser frequency, but to stabilize the frequency of the master microwave generator!

Different methods (polarimetric, saturated absorbtion, etc.) can be used to stabilize the laser frequency in CPT atomic clocks. But initially [1] the method identical to one “proposed” by authors of the article under review was used – namely, method of slow modulation of the laser current with subsequent lock-in detection of the intensity of light that has passed through the working cell.

Thus, in the pioneering work [1] we read:

“The laser frequency was locked to the peak of this absorption profile by modulating the laser current at 10 kHz and using lock-in detection as shown in Fig. 2. The resulting laser frequency excursion was roughly 10 MHz. The detuning from the exact line center could be varied by using a voltage offset at the integrator input.

The 4.6 GHz RF modulation applied to the laser was synthesized from a 5 MHz precision crystal oscillator. When the RF modulation frequency was exactly equal to one half the hyperfine splitting of the Cs ground state, the sideband absorption peak height was reduced by 0.5% due to the CPT effect”.

Thus, slow (10 kHz) modulation of the laser current was used to stabilize the laser frequency, and fast modulation was used to stabilize the microwave generator. As far as can be seen, the method used by Kitching et al. in 2000 was no different from the method described by the authors of the article under review.

In terms of laser frequency stabilization, chip-scale atomic clocks, magnetometers, and NMR gyroscopes are no different, so the use of a standard laser stabilization method in a gyroscope is not new. Moreover, the method described by the authors is used in existing NMR gyroscope projects due to its simplicity and standardity.

Authors  declare:

“Our primary aim is to facilitate the pumping of Cs atoms, and this method does not rely on the CPT effect”.

One can be absolutely sure that none of existing NMR gyroscope projects “rely on the CPT effect” due to the absence of a microwave field source in the gyroscope scheme.

Thus, I believe that the article in a new version can be published if the authors renounce claims to the novelty of the method of stabilizing laser radiation. If they are not ready to do this, they must clearly indicate the fundamental differences of their method from the methods described in the literature, in particular, in [1] as well as in article cited in my previous review. But I categorically advise against doing this, since I do not see any fundamental differences.

Next, authors need to back up their claim:

L17: In addition, a laser 16 frequency stabilization method based on injection current modulation is introduced, which can theoretically achieve wavelength fluctuations of less than 3×10-6 nm within 180 s.

Theoretical estimates should be based on the optical transition line width and the ratio of laser power to noise power, including shot noise. I only saw experimental estimates in the text (L280).

I would also like to ask the authors to use the appropriate units (e.g. gigahertz) wherever they talk about optical line widths, laser stability, etc., and not compare gigahertz with nanometers.

REFERENCES

1.Kitching, John, et al. "A microwave frequency reference based on VCSEL-driven dark line resonances in Cs vapor." IEEE Transactions on Instrumentation and Measurement 49.6 (2000): 1313-1317

Author Response

We fully agree with your point of view and have made the necessary modifications according to your suggestions. Please see the attachment.
